# Recurrence rate of preterm birth and associated factors among women who delivered at Kilimanjaro Christian Medical Centre in Northern Tanzania: A registry based cohort study

**Nathaniel Halide Kalengo**[1]*, **Leah A. Sanga**[1], **Rune Nathaniel Philemon**[1], **Joseph Obure**[2], **Michael J. Mahande**[1]

1 Department of Epidemiology & Biostatistics, Institute of Public Health, Kilimanjaro Christian Medical College, Moshi, Tanzania, 2 Department of Obstetrics and Gynaecology, Kilimanjaro Christian Medical Centre, Moshi, Tanzania

* nathanhalide@yahoo.com

**Data Availability Statement:** The KCMC medical birth registry data contains potentially identifying and sensitive patient information. This has also

## Abstract

### Background

Preterm birth is a public health problem particularly in low- and middle-income countries especially in sub-Saharan Africa. It is associated with infant morbidity and mortality. Survivor of preterm suffers long term health consequences such as respiratory, hearing and visual problems as well as delivering preterm infants. Preterm birth also tends to recur in subsequent pregnancies. Little is known about recurrent rate of preterm birth and associated factors in Tanzania. This study aimed to determine the recurrence rate of preterm birth and associated factors among women who delivered at Kilimanjaro Christian Medical Centre (KCMC), in Northern Tanzania.

### Methods

A historic cohort study was designed using maternally-linked data from KCMC medical birth registry. Women who delivered 2 or more singletons were included. A total of 5,946 deliveries were analysed. Recurrence of preterm birth and associated risk factors were estimated using multivariable log-binomial regression model with robust standard error to account for repeated births from the same mother.

### Results

Overall recurrent rate of preterm birth was 24.4%. The recurrence of early preterm birth was higher compared to late preterm birth (26.2% vs. 24.2%). Similar pattern of recurrence was observed for spontaneous and medically indicated preterm birth (13.5% vs. 10.9%, respectively). Previous preterm birth (RR;1.85, 95% CI: 1.49, 2.31), preeclampsia (RR;1.46, 95%

been stipulated by the local Institutional Review Board of KCMC hospital and the National Ethics Committee in Norway when establishing this birth registry. Permission to use the data in this study was made through the Kilimanjaro Christian Medical University College Research and Ethics Review Committee, and received an approval number 2086. The authors do not have the legal right to share the data publicly. All data requests can be sent directly to the Executive Director of the KCMC referral hospital, P. O. Box 3010, Moshi, Tanzania, Email: kcmcadmin@kcmc.ac.tz.

**Funding:** This study was supported through the DELTAS Africa Initiative Sub-Saharan Africa Consortium for Advanced Biostatistics (SSACAB) (Grant No. 107754/Z/15/Z). The DELTAS Africa Initiative is an independent funding scheme of the African Academy of Sciences (AAS) Alliance for Accelerating Excellence in Science in Africa (AESA) and is supported by the New Partnership for Africa's Development Planning and Coordinating Agency (NEPAD Agency) with funding from the Wellcome Trust (Grant No. 107754/Z/15/Z) and the UK government. The views expressed in this publication are those of the authors and not necessarily those of the AAS, NEPAD Agency, Wellcome Trust or the UK government.

**Competing interests:** The authors have declared that no competing interests exist.

CI: 1.07, 2.00), long inter-pregnancy interval (RR;1.22, 95% CI: 1.01, 1.49) and clinical sub-types (RR = 1.37, 95% CI: 1.00, 1.86) were important predictors for recurrent preterm birth.

## Conclusion

Recurrence of preterm birth remains higher in this population. The rate of preterm recurrence was dependent of gestational age and sub-clinical subtype. Other factors which were associated with recurrence of preterm birth were previous preterm birth, preeclampsia and long inter-pregnancy interval. Early identification of high risk women during prenatal period is warranted.

## Introduction

Globally, approximately 15 million babies are born preterm annually [1]. Majority (>60%) of preterm births occur in low and mid-income countries particularly in South Asia and sub-Saharan Africa [2]. More than 1 million babies who are born preterm die annually due to pre-term related complications while those who survive suffers long-term lifetime consequences such as learning disability, visual and hearing problems [3]. Furthermore, preterm birth (PTB) is a leading cause of neonatal morbidity and mortality [4, 5]. It accounts for 35% of neonatal mortality globally each year [4]. The severity of PTB differs across the gestational age [6]. A systematic analysis from 171 countries in low and high income countries reported the PTB rate ranging from 5% to 18% [4]. In Tanzania, approximately 2 million births occur annually, where 11% of these babies are born preterm [7]. In Tanzania, PTB has been reported to account for 10% of under five mortality [7].

Previous investigators have demonstrated that women who had previous history of PTB are at higher risk of experiencing PTB in their subsequent pregnancies [8]. The authors noted that the recurrence can be as high as 30% [8]. Similarly, a hospital-based study in Tanzania reported PTB recurrence rate of 17%, in this study the recurrence of PTB accounted for 15% of perinatal mortality [9]. The recurrence of preterm birth is dependent on the gestational age of the index pregnancy, it also vary by clinical subtype. The shorter gestation age in the index pregnancy has also been reported to increase the risk of PTB in the subsequent pregnancy [10]. Previous investigators have showed that 70% of preterm births occur spontaneously [8] while 30% are medically indicated preterm birth [11].

Numerous factors have been associated with an increased risk of PTB and its recurrence. These include history of preterm birth, low maternal body mass index, pre-eclampsia, placenta previa, and short cervical length [12, 13]. Furthermore, genital infection, multiple gestation, short interpregnancy interval, uterine anomalies, smoking, black race, raised fetal fibronection concentration, psychological stress, stressful working condition, poor living condition, low level of education and maternal unemployment status have aso been associated with an increased risk of PTB and its recurrence [14–16].

Previous studies in the developed countries have reported that the recurrence risk of PTB varies according to gestational age of the previous pregnancy [6]. The authors noted that recur-rent PTB also affect the survival of preterm in the successive pregnancy. However, a previous study in Tanzania using the same data used by the present study did not assess the effect of ges-tational age category of the index pregnancy and its implications on recurrent preterm birth and perinatal outcomes (Mahande *et al.*, 2013). This might have underestimated the rate of PTB recurrence and associated perinatal outcomes. Furthermore, some authors have also

reported that the recurrence risk of PTB varies by clinical-subtype [11, 17]. This also have not been explored in resource setting including Tanzania. This study aimed to determine the recurrence rate of PTB across gestational age of the index pregnancy and according to clinical subtypes among women who delivered at Kilimanjaro Christian Medical Center. The study also explored the factors associated with recurrence of PTB across gestational age and by clinical subtype.

Understanding the recurrence rate of PTB is critical as it provides information to clinicians to guide clinical counselling for women at risk of recurrent preterm birth. Information on risk factors for recurrence of PTB are important to researchers as it suggest the etiological factors for PTB and thus guide development of focused interventions to prevent recurrent PTB and its associated consequences. Averting the perinatal mortality attributed by PTB and its complications may help to accelerate the achievement towards Sustainable Development Goal 3.2 which aims to end preventable deaths of new-borns and children under 5 years, with all countries targeting to reduce neonatal mortality to at least as low as 12/1000 live births and under 5 mortality to at least as low as 25/1000 live births by 2030.

## Methods

### Study setting and design

This was a hospital-based retrospective cohort study which was designed using maternally-linked data from KCMC Medical Birth Registry for women with consecutive deliveries between 2000 and 2015. The KCMC hospital is located in Moshi, northern Tanzania. KCMC is one of four referral hospital in Tanzania serving more than 1.64 million people [18]. Approximately 4,000 deliveries are recorded annually. The Obstetrics and Gynaecology Department at KCMC receives 80% of patients from Moshi who are self-referred which is the main catchment area, while 20% of patients are referred from other nearby regions. KCMC Medical Birth Registry was established in 2000 by University of Bergen Norway in collaboration with Kilimanjaro Christian Medical University College.

### Study population and sample size

We included all women who delivered singleton for least two consecutive births at KCMC between 2000 and 2015. All births with missing gestational age, gestational age less than 28 weeks gestational age, multiple pregnancies and those who were referred from rural due to medical complications were excluded. The final sample size comprised of 5,946 consecutive singleton deliveries.

### Data collection

The birth registry recorded information for all mothers who delivered in the obstetrics and gynaecology department at KCMC. Trained midwives carry out daily interviews using a standardized questionnaire within 24 hours of delivery, or as soon as a mother has recovered. Other relevant information was obtained from antenatal cards and medical records. All data were stored in a computerized database system at the medical birth registry office.

### Variable definitions

The outcome of interest was recurrence of preterm birth. The recurrent preterm birth was defined as two or more deliveries of live babies before 37 completed weeks of gestation [8, 19]. The preterm birth was categorized based on gestation age (early or late PTB). Early preterm birth was defined as a birth of a live baby between 28 and < 32 weeks of gestation age while

late preterm birth was referred to birth of a live baby between 32 and <37 weeks of gestation age. Furthermore, the preterm birth was also categorized according to clinical subtypes (i.e. occurred spontaneously or due to medical indication).

The primary exposure of interest preterm birth in the index pregnancy. Other covariates included birth weight, pre-eclampsia, maternal age, maternal education level, residence, occupation, alcohol use in pregnancy, clinical subtypes, inter-pregnancy interval (IPI) and premature rupture of membranes in the index pregnancy. Inter-pregnancy interval was recorded as continuous then categorized into normal IPI (24 to 59 months), short IPI (<24 months) and long IPI (>59 months).

### Data analysis

Data cleaning and analysis was done using STATA version 13.1 (Stata Corp, College Station, TX, USA). Continuous variables were summarized using measures of central tendency with their respective measure of dispersion while categorical variables were summarized using frequencies and percentages. Using births as unit of analysis, multivariable log-binomial regression model with robust standard error was used to estimate the recurrence rate of PTB with their 95% confidence intervals while considering the repeated observations (births) from the same mother. All variables which were significant at 20% in bivariate analysis were included in multivariable analysis. In multivariable model, previous preterm was treated as the main exposure of interest. The risk ratios was calculated to find the association of the outcome of interest and the exposures under study. Akaike information criteria (AIC) was used to select the best model in a backwards elimination. A P-value of less than 0.05 was considered significance.

### Ethical considerations

The study was approved by Kilimanjaro Christian Medical University College Research and Ethics Committee prior to commencement of the study. The informed verbal consents were obtained from each participant before recruitment. For women who did not consent to participate in the study were not included. The confidentiality and privacy were assured by the use of mothers' identification numbers and private rooms for interviews.

## Results

### Socio-demographic characteristics of the study participants

A total of 5,946 consecutive singleton deliveries were analysed (Fig 1). The mean age of the mother's was 30.4 (SD = 5.1) years. Majority (64.7%) were aged between 25–35 years. More than a third (68.3%) were from urban and. more than a quarter (29.4%) reported using alcohol during pregnancy (Table 1).

### Clinical characteristics of the study participants

Only 182 (3.1%) reported to have pre-eclampsia and 727 (12.2%) had malaria during pregnancy. Majority (87.8%) of women had normal birth weight. Sixty eight percent of women had spontaneous vaginal delivery (68.15%) while the remaining had medically indicated deliveries. Nearly half (47.2%) of the women had optimal inter pregnancy interval. The majority (85.7%) had a parity of two to five. A third (63.1%) of the women had more than 4 antenatal care visits (Table 2).

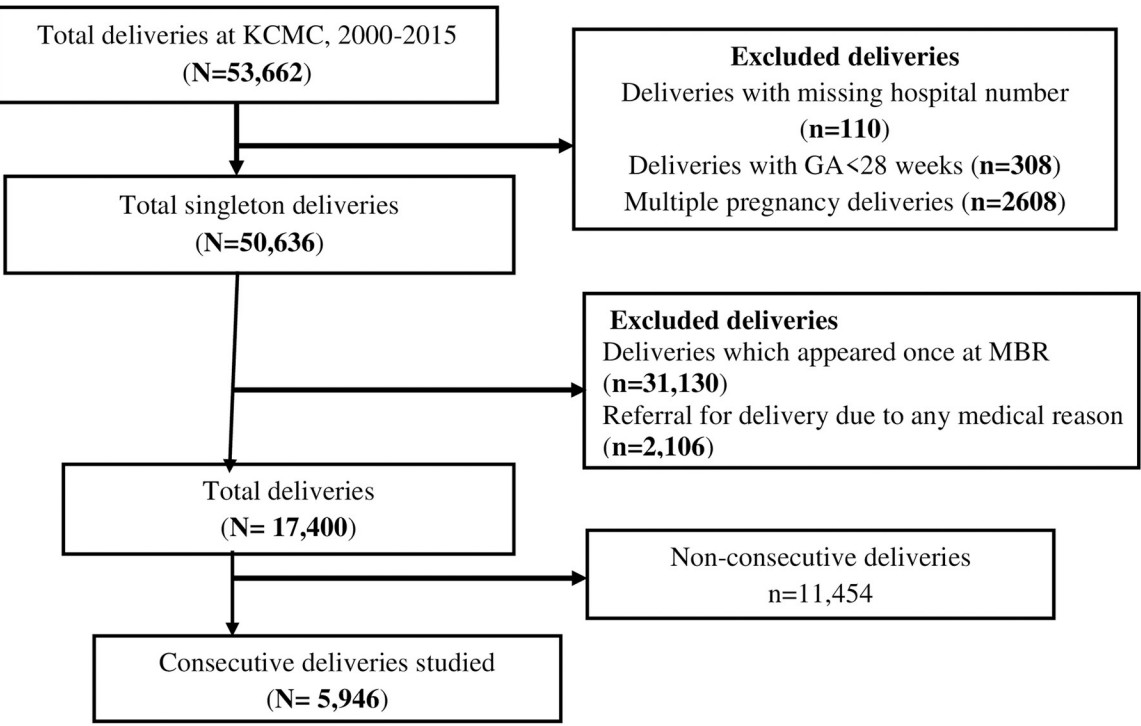

**Fig 1. Flow chart of the number of deliveries, inclusions and exclusions during the study period.**

## Recurrence of preterm birth

The overall rate of recurrence of preterm birth in this population was 24.4%. The rate of recurrence of early preterm birth was slightly higher compared to late preterm birth (26.2% vs. 24.2%, respectively). In the early preterm birth category, 20% of deliveries occurred spontaneously while 2.2% were medically indicated. In late preterm birth category, 16.6% of deliveries occurred spontaneously while 5.1% were medically indicated (Table 3).

On the other hand, the recurrence rate of spontaneous preterm birth was higher compared to medically indicated preterm birth (13.5% vs 10.9%, respectively). The rate of spontaneous preterm birth in previous preterm which recurred spontaneously was 8.3%. The rate of spontaneous preterm birth in prior pregnancy which recurred due medical indication was 2.1%. The rate of medically indicated preterm birth which recurred spontaneously was 2.8% while the rate of medically indicated preterm birth in prior pregnancy which recurred due to medical indication was 10.3% (Table 3).

The rate of early medically indicated preterm birth in previous pregnancy which recurred spontaneously was 6.3% while 31.3% recurred due to medical indication. The rate of late medically indicated preterm birth which recurred spontaneously was 6.7% while 22.7% recurred due to medical indications (Table 3).

In bivariate analysis (Table 4), the factors associated with recurrent preterm birth which were statistically significant at 20% level were previous preterm birth, pre-eclampsia, inter-pregnancy interval, birth weight, clinical subtypes and alcohol use. After adjusting for mother's age, mother's education, premature rupture of membrane and alcohol use, factors which were statistically significant were previous preterm birth, pre-eclampsia, inter-pregnancy interval and clinical subtypes.

**Table 1. Socio-demographic characteristics of women with consecutive deliveries (N = 5,946).**

| Variables | n | % |
|---|---|---|
| Mother's age (years) | | |
| 15-<25 | 786 | 13.2 |
| 25–35 | 3,848 | 64.7 |
| >35 | 1,303 | 21.9 |
| Missing | 9 | 0.2 |
| Mother's residence | | |
| Rural | 1,640 | 27.6 |
| Urban | 4,304 | 68.3 |
| Missing | 2 | 0 |
| Marital status | | |
| Not married | 185 | 3.1 |
| Married | 5,761 | 96.9 |
| Educational level of mother | | |
| None | 56 | 0.9 |
| Primary | 2,880 | 48.4 |
| Secondary | 872 | 14.7 |
| Tertiary | 2,138 | 36 |
| Mother's occupation | | |
| Employed | 4,486 | 75.4 |
| Unemployed | 1,445 | 24.3 |
| Missing | 15 | 0.3 |
| Alcohol use | | |
| No | 4,196 | 70.6 |
| Yes | 1,750 | 29.4 |
| Religion | | |
| Catholic | 2,526 | 42.5 |
| Protestant | 2,385 | 40.1 |
| Muslim | 1,008 | 17 |
| Others | 17 | 0.3 |
| Missing | 10 | 0.2 |
| Mother's tribe | | |
| Chagga | 3,864 | 65 |
| Pare | 720 | 12.1 |
| Others | 1,362 | 22.9 |

The bivariate analysis for the factors associated with recurrent preterm birth has been shown in Table 5. Women who had previous preterm birth had nearly 2-fold risk of preterm birth in their subsequent pregnancies compared to women who had term pregnancies in the previous pregnancies (RR = 1.85; 95% CI: 1.49-, 2.31). Women who had preeclampsia in previous pregnancies had 50% (RR = 1.46; 95% CI: 1.07, 2.00) higher risk of preterm birth in their subsequent pregnancies as compared to women who had normotensive pregnancy in their previous pregnancy. Women who had short IPI in the previous pregnancy had 1.13 times (95% CI: 0.96, 1.33) higher risk of having future preterm birth compared to women who had optimal IPI whereas women who had long IPI had 22% (RR = 1.22, 95% CI: 1.01, 1.49) higher risk of having preterm birth in their future pregnancies as compared to their counterparts who had optimal IPI.

**Table 2. Clinical characteristics of mothers with consecutive deliveries at KCMC (N = 5,946).**

| Variables | n | % |
|---|---|---|
| Pre-eclampsia | | |
| **No** | 5,764 | 96.9 |
| **Yes** | 182 | 3.1 |
| Birth weight | | |
| **Normal birth weight** | 5,542 | 93.2 |
| **Low birth weight** | 359 | 6 |
| **Very low birth weight** | 37 | 0.6 |
| **Extremely low birth weight** | 8 | 0.1 |
| Malaria during pregnancy | | |
| **No** | 5,219 | 87.8 |
| **Yes** | 727 | 12.2 |
| Clinical subtypes | | |
| **Spontaneous delivery** | 4,052 | 68.15 |
| **Medically indicated delivery** | 1,884 | 31.69 |
| **Missing** | 10 | 0.17 |
| Inter-pregnancy interval (months) | | |
| **24–59** | 2,805 | 47.2 |
| **<24** | 1,902 | 32 |
| **>59** | 1,049 | 17.6 |
| **Missing** | 190 | 3.2 |
| Parity | | |
| **Prime** | 734 | 12.3 |
| **Multiparity** | 5,093 | 85.7 |
| **Grand multiparity** | 119 | 2 |
| Antenatal care attendance | | |
| **4 and more visits** | 3,751 | 63.1 |
| **Less than 4 visits** | 2,139 | 36 |
| **Missing** | 56 | 0.9 |

Women who delivered by medical indication had 37% (RR = 1.37, 95% CI: 1.00, 1.86) higher risk of delivering preterm birth in their subsequent pregnancies compared to those who had spontaneous preterm deliveries. Furthermore, women who had term deliveries had 41% lower risk of having preterm deliveries in their subsequent deliveries compared to women who delivered spontaneously (RR = 0.59, 95% CI: 0.46, 0.77). Women whose babies had low birth weight in their previous pregnancy had 1.57 times (95% CI: 1.25, 1.99) higher risk of delivering preterm babies in the subsequent pregnancies compared to women who had babies with normal birth weight. Women whose babies had very low birth weight had 2 times (95% CI: 1.29, 3.10) higher risk of preterm birth recurrence compared to women whose babies had normal birth weight. In the other hand, women whose babies had extremely low birth weight had 1.13 times (95% CI: 0.31, 4.10) higher risk of preterm birth recurrence compared to those with babies having normal birth weight.

## Discussion

This study aimed to determine the recurrence of preterm birth and it associated factors among women with subsequent births. The overall recurrence rate of preterm birth was 24.4%. The recurrence of early preterm birth was slightly higher compared to the late preterm birth

**Table 3. Classification of rate of preterm birth recurrence by clinical subtypes and gestational age categories (N = 5946).**

|  | | | Current preterm birth categories | | |
|---|---|---|---|---|---|
|  | Total | Term | Preterm | sPTB | mPTB |
|  | N | n (%) | n (%) | n (%) | n (%) |
| Previous Preterm deliveries | | | | | |
| **Term delivery** | 5,405 | 4880 (90.3) | 525 (9.7) | 330 (6.1) | 195 (3.6) |
| **Preterm delivery** | 541 | 409 (75.6) | 132 (24.4) | 73 (13.5) | 59 (10.9) |
| **Spontaneous delivery** | 4,309 | 3863 (89.6) | 446 (10.4) | 358 (8.3) | 88 (2.1) |
| **Medically indicated delivery** | 1,619 | 1408 (87.9) | 211 (13.1) | 45 (2.8) | 166 (10.3) |
| Early preterm birth | 61 | 45 (73.8) | 16 (26.2) | 10 (16.4) | 6 (9.8) |
| **Spontaneous delivery** | 45 | 35 (77.8) | 10 (22.2) | 9 (20) | 1 (2.2) |
| **Medically indicated delivery** | 16 | 10 (62.5) | 6 (37.6) | 1 (6.3) | 5 (31.3) |
| Late preterm birth | 480 | 364 (75.8) | 116 (24.2) | 63 (13.1) | 53 (11) |
| **Spontaneous delivery** | 314 | 246 (78.3) | 68 (21.7) | 52 (16.6) | 16 (5.1) |
| Medically indicated delivery | 163 | 115 (70.6) | 48 (29.4) | 11 (6.7) | 37 (22.7) |

(26.4% vs. 24.2%, respectively). Similarly, the recurrence of spontaneous preterm birth was higher compared to medically indicated preterm birth (13.5% vs. 10.9%, respectively). The previous preterm birth, inter-pregnancy interval, preeclampsia were significantly associated with recurrence of preterm birth.

The rate of preterm birth recurrence of 24.4% observed in the present study is higher than the recurrence rate of 17% which was previously reported in 2008 using the same dataset [9]. This reflect that the rate of preterm birth recurrence has increased by 43.5% percentage points over the last seven years. This increase may due increased medical induced deliveries. This calls for a need for concerted effort to address the risk factors associated with preterm birth in order to reduce this rising recurrence of preterm birth. The previous study in Canada reported a preterm birth recurrence rate of 30% [8] which is higher compared to the findings of this study. These differences may be explained by the difference in study designs between the two studies. The former was a meta-analysis, which reported over 30 clinical trials and cohort findings. Probably this might have more precision and high quality data than the present study. Another possible explanation could be that the recurrence rate of preterm birth is higher in low resourced countries than in high resource settings, but it looks lower due to poor data quality and a huge number of unreported cases. However, despite of having slightly lower rate of recurrence, the survival gap for preterm babies is high between the two settings with majority (90%) of all preterm babies die in low resourced settings as compared to their counterparts born in developed world where only 10% of preterm babies die [1].

We found a differential in preterm birth recurrence by gestational age category where 26.2% of recurrence occurred in early preterm. This information is important because babies at this gestational account many challenges in terms of management and care especially in low resource settings due to unavailability of high impact interventions known to serve lives for preterm babies. A study done in Canada reported that gestation age is inversely proportion to increased morbidity and mortality [8]. Therefore, special care is needed for babies in this category especially in our settings where there is still a challenge in caring of early preterm deliveries. 24.2% of late preterm birth recurred at late preterm birth category.

There was a variation in recurrence of preterm birth by clinical subtypes and across gestational categories. The recurrence rate of spontaneous preterm birth was higher than the recurrence of medically indicated (13.5% vs. 10.9%, respectively). However, the recurrence of spontaneous preterm birth in our study was lower compared to 70% that was reported

**Table 4. Factors associated with recurrent preterm births among women delivered at KCMC (N = 5946).**

| Characteristics | Total (N) | Recurrence of Preterm birth | | |
|---|---|---|---|---|
| | | n (%) | cRR,95% CI | P-value |
| in previous birth) | | | | |
| **Term delivery** | 5,405 | 525 (9.7) | 1 | |
| **Preterm delivery** | 541 | 132(24.4) | 2.51[2.11,2.97] | <0.001 |
| Pre-eclampsia | | | | |
| **No** | 5,763 | 622(10.8) | 1 | |
| **Yes** | 183 | 35(19.1) | 1.77[1.29,2.41] | <0.001 |
| Inter-pregnancy interval (months) | | | | |
| **24–59** | 2,805 | 281(10) | 1 | |
| **<24** | 1,902 | 227(11.9) | 1.19[1.01,1.40] | 0.036 |
| **>59** | 1,049 | 130(12.4) | 1.24[1.02,1.51] | 0.034 |
| Birth weight | | | | |
| **Normal birth weight** | 5,441 | 538(9.9) | 1 | |
| **Low birth weight** | 442 | 98(22.2) | 2.24[1.85,2.71] | <0.001 |
| **Very low birth weight** | 53 | 19(35.8) | 3.62[2.51,5.21] | 0.001 |
| **Extremely low birth weight** | 10 | 2(20) | 2.02[0.58,6.99] | 0.267 |
| Premature rupture of membrane | | | | |
| **No** | 5,873 | 642(10.9) | 1 | |
| **Yes** | 73 | 15(20.5) | 1.00[0.57,1.76] | 0.988 |
| Clinical subtypes | | | | |
| **Spontaneous delivery** | 359 | 78(21.7) | 1 | |
| **Medically indicated delivery** | 178 | 54(30.3) | 1.40[1.03,1.88] | 0.028 |
| **Term delivery** | 5409 | 527(9.74) | 0.45[0.36,0.57] | <0.001 |
| Mother's education | | | | |
| **No formal education** | 56 | 7(12.5) | 1 | |
| **Primary education** | 2,880 | 350(12.2) | 0.98[0.49,1.96] | 0.945 |
| **Secondary education** | 872 | 105(12) | 0.96[0.47,1.96] | 0.906 |
| **Tertiary education** | 2,138 | 195(9.1) | 0.73[0.36,1.48] | 0.384 |
| Alcohol use | | | | |
| **No** | 4,196 | 485(11.6) | 1 | |
| **Yes** | 1,750 | 172(9.8) | 0.85[0.72,1.00] | 0.055 |
| Mother's age (years) | | | | |
| **15-<25** | 786 | 100(12.7) | 1 | |
| **25–35** | 3,848 | 404(10.5) | 0.82[0.67,1.01] | 0.059 |
| **>35** | 1,303 | 152(11.7) | 0.92[0.72,1.16] | 0.46 |

cRR: Crude risk ratio

elsewhere [8]. This difference could be explained by the differences in population characteristics between the studies populations. Medically indicated preterm birth have been a reason for the rise of preterm birth in the US whereby it contributes up to 30% of all preterm births [11]. In the presented study, medically indicated preterm births accounted for 44.6% of all recurrent preterm birth in the study setting. These calls for targeted intervention to prevent medically indicated preterm births thereby helping to reduce the overall preterm birth rate and associated cost implications.

Previous preterm birth was the main reason for recurrence of preterm birth in our hospital. Women who had previous preterm birth were at higher risk of recurrence compared to

**Table 5. Factors for recurrence of preterm birth delivery by women characteristics in previous pregnancy (N = 5,946).**

| Variables | cRR,95% CI | aRR,95% CI | P-value |
|---|---|---|---|
| Birth type | | | |
| **Term delivery** | 1 | 1 | |
| **Preterm delivery** | 2.51[2.11,2.97] | 1.85[1.49,2.31] | <0.001 |
| Pre-eclampsia | | | |
| **No** | 1 | 1 | |
| **Yes** | 1.77[1.29,2.41] | 1.46[1.07,2.00] | 0.017 |
| Inter-pregnancy interval | | | |
| **Normal IPI** | 1 | 1 | |
| **Short IPI** | 1.19[1.01,1.40] | 1.13[0.96,1.33] | 0.148 |
| **Long IPI** | 1.24[1.02,1.51] | 1.22[1.01,1.49] | 0.042 |
| Clinical subtypes | | | |
| **Spontaneous delivery** | 1 | 1 | |
| **Medically indicated** | 1.40[1.03,1.88] | 1.37[1.00,1.86] | 0.048 |
| **Term delivery** | 0.45[0.36,0.57] | 0.59[0.46,0.77] | <0.001 |

cRR: Crude risk ratio, aRR: Adjusted risk ratio

women who had term delivery in their previous pregnancies. These findings were similar to the study done in Canada which reported that previous preterm was the main risk for future preterm birth from the same mother [8]. This calls for a need for early identification of women at risk of recurrence through counselling about the syndrome and for possible management and close follow-up during the prenatal care.

Other factors which were associated with recurrence of preterm birth include medically indicated preterm birth, preeclampsia and long IPI. Medically indicated preterm birth were having higher risk of recurring in subsequent pregnancies compared to spontaneous deliveries. Also, this study found that term deliveries were less likely to have siblings who will be born before term either spontaneously or medically indicated. This is consistent with previous studies [11, 17, 20]. Authors in the previous studies suggested that the clinical subtypes share the etiologies which could explain the observed results.

Women with preeclampsia in previous pregnancy had higher risk preterm birth in their subsequent pregnancies. This finding is in agreement with previous investigators in developed countries [17]. Proper preeclampsia management is vital in averting the recurring preterm birth due to this factor. Long IPI was found to be associated with recurrence of preterm birth. Our finding was in congruent with previous studies in northern Tanzania [21]. This suggests a need for use of family planning methods to ensure optimal IPI thereby averting preterm birth attributed to long IPI. Women should be educated on the risk of long IPI on preterm birth during ANC visits.

## Strengths and limitations

The strength of this study is the use of large sample of pregnant women who delivered consecutively at KCMC for the past 15 years. This gave a high power of a study to detect the association of different factors with recurrence of preterm birth. This data accounted for clustering effect with robust variance since consecutive deliveries of the same mothers were studied. To the best of author's knowledge, this is the first study to describe recurrence of preterm birth by clinical subtypes and by gestational age categories in Tanzania.

However, this study had some limitations which need to be taken into accounting when interpreting our finding. Use of secondary data lead to missing some variables which are risk for preterm birth and possibly its recurrence such as paternal change between successive pregnancies and behavioural factors.

## Conclusion

Recurrence of preterm birth is still a major public health problem. The gestation age at delivery is a predictor of the morbidity and mortality due to preterm births recurrence. The factors which were associated with recurrence of preterm birth were previous preterm birth, pre-eclampsia, long inter-pregnancy interval, and clinical subtypes of preterm birth. Women with history of preterm birth should be identified as high risk group and counselling should be done with regards to the recurrence of preterm birth.

## Supporting information

**S1 Questionnaire.**
(DOCX)

## Author Contributions

**Conceptualization:** Nathaniel Halide Kalengo, Rune Nathaniel Philemon, Joseph Obure, Michael J. Mahande.

**Data curation:** Nathaniel Halide Kalengo, Leah A. Sanga.

**Formal analysis:** Nathaniel Halide Kalengo.

**Methodology:** Nathaniel Halide Kalengo, Michael J. Mahande.

**Writing – original draft:** Nathaniel Halide Kalengo.

**Writing – review & editing:** Nathaniel Halide Kalengo, Leah A. Sanga.

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
