## [Decision Letter · Decision Letter 0]

6 May 2020

PONE-D-19-30288

Recurrence rate of preterm birth and associated factors among women who delivered at Kilimanjaro Christian Medical centre in Northern Tanzania: a registry based cohort study

PLOS ONE

Dear Dr. Kalengo,

Thank you for submitting your manuscript to PLOS ONE. After careful consideration, we feel that it has merit but does not fully meet PLOS ONE’s publication criteria as it currently stands. Therefore, we invite you to submit a revised version of the manuscript that addresses the points raised during the review process.

We would appreciate receiving your revised manuscript by Jun 20 2020 11:59PM. To enhance the reproducibility of your results, we recommend that if applicable you deposit your laboratory protocols in protocols.io, where a protocol can be assigned its own identifier (DOI) such that it can be cited independently in the future. For instructions see: http://journals.plos.org/plosone/s/submission-guidelines#loc-laboratory-protocols

We look forward to receiving your revised manuscript.

Kind regards,

Joel Msafiri Francis, MD, MS, PhD

Academic Editor

PLOS ONE

Journal Requirements:

2.  Please include additional information regarding the survey or questionnaire used in the study and ensure that you have provided sufficient details that others could replicate the analyses. For instance, if you developed a questionnaire as part of this study and it is not under a copyright more restrictive than CC-BY, please include a copy, in both the original language and English, as Supporting Information.  If the original language is written in non-Latin characters, for example Amharic, Chinese, or Korean, please use a file format that ensures these characters are visible.

3. Please amend your current ethics statement to address the following concerns:  

a) Did participants provide their written or verbal informed consent to participate in this study?

Additional Editor Comments (if provided):

Reviewers' comments:

Reviewer's Responses to Questions

**Comments to the Author**

1. Is the manuscript technically sound, and do the data support the conclusions?

Reviewer #1: Yes

Reviewer #2: Partly

Reviewer #3: Yes

2. Has the statistical analysis been performed appropriately and rigorously? 

Reviewer #1: Yes

Reviewer #2: Yes

Reviewer #3: Yes

3. Have the authors made all data underlying the findings in their manuscript fully available?

Reviewer #1: Yes

Reviewer #2: Yes

Reviewer #3: No

4. Is the manuscript presented in an intelligible fashion and written in standard English?

Reviewer #1: Yes

Reviewer #2: Yes

Reviewer #3: Yes

5. Review Comments to the Author

Reviewer #1: Preterm birth is an important medical problem needing academic efforts. This work was very well done, however, I did not learn anything by reading the manuscript. All the results are as expected and no new discovery would benefit the research and management of preterm birth. It is not challenging academically, therefore, I suggest the rejection of the work.

Reviewer #2: Overall:

It’s an important study as preterm labour and delivery is an important cause of perinatal deaths and long term complications in the newborns. This information will benefit the studied community as the gained information will be used in managing future pregnancies of patients in this locality and Tanzania in general.

General comment:

There are few grammatical errors which need to be corrected.

Title: Well written, clear, and in line with the objectives. It explains what has been researched.

Abstract: The abstract is provided and it contains all the important elements of the study.

Introduction:

The subject has been well researched. There is mixed information about the concept of risk and recurrence in second and third paragraphs, which I feel should be separated.

Last but one paragraph shows that the “study will be done”. This needs to be written in past tense.

Methods:

The methods have been explained, and have provided answers to raised questions, but do not explain whether this was a prospective or retrospective study. This could explain the data that was missing.

The category for the late preterm birth as explained it the first paragraph of variable definitions is “<37 weeks of gestation age, which means 36 completed weeks. This means there are women who were missed who were in the 37th week and had not attained 37 completed weeks

Ethical considerations:

It was ethical to carry out this study as it caused no reported harm to the participants.

Ethical issues were well treated as women gave informed consent, even though it is not explained whether there were women who did not give consent and what happened to them.

Results:

The results section is provided, and explains the results well.

There is too much text that explains the results while the tables are provided. The narrative part can concentrate on highlighting only important findings in the table.

Some sentences are not very clear, and few are hanging.

Discussion:

This is provided and has discussed the results of their study but it does not discuss very well the results and link them well with other previous studies.

Conclusion:

Conclusions have been given. It’s important to limit the conclusions to only what has been studied.

Reviewer #3: Thank you for the opportunity to review this manuscript of an important topic in low income country setting.

Here are my comments to improve the manuscript.

ABSTRACT:

The conclusion misses some of the important variables that were significant Need to mention the numerous factors associated with the recurrence.

INTRODUCTION

Please put the long form of KCMC on page 4 paragraph 1 because it is used for the first time.

Write Sustainable Development Goals and not sustainable development goals on page 4 paragraph 2

METHODOLOGY

Study population and study sample: It is good practice to show power calculation to see whether the sample size in the database is enough to depict the outcome intended.

Why were the referred cases with medical complications from rural area excluded. Some could have been referred due to conditions that needed medically induced delivery in a tertiary institution.

Variable definitions: There are other important conditions associated with PTb such as diabetes mellitus, chronic hypertension. But these were not factored in. Can you explain if these conditions are not captured in the database.

Data Analysis: You should add a description about risk ratio in the text.

Ethical considerations: Write long form for KCMUCo

RESULTS

Page 6 last paragraph:"Through spontaneous vaginal delivery (68.15%)". This sentence is hanging it is not clear.

Table 1: Please check the age range 15 -25 and 25 -35. It seems 25 years are included into the two groups which is misleading in my opinion

Table2: Please rearrange the IPI starting with <24 then 24-29 etc. On parity how do you have primigravida if the inclusion criteria was two or more singleton deliveries: The number of antenatal visits: what about those with four visits it seems they are missing here.

Table 3: A bit confusing as the row numbers add more than the total. May be you need to remove the column with TOTAL N.

Recurrence of preterm birth ; I would suggest you put a numerator and denominator for there overall recurrent rate of 24.4% as it not very clear from table 3.

DISCUSSION

Page 8 last paragraph:Why do you think it has increased. Is because of referral cases or because of the population increase or because of increase in medically induced deliveries?

Pg 9 Paragraph 1. You can not really make comparison using the meta analysis in my opinion. Rather may be look at the individual papers in meta analysis to make a comparison.

Pg 9 Paragraph 3:For the medically indicated preterm of 44.6%, Probably should say that for medically induced, preterm care should be improved to make sure the premature babies are taken care properly rather than need to be avoided as you can not avoid if delivery is indicated before term.

Paragraph 3 page 10: I think family planning is important but those with long IPI probably were using FP or thee was delay because of other reasons. I think here what is important is to educate the women on the risks of long IPI on preterm birth during the antenatal period.

6. PLOS authors have the option to publish the peer review history of their article (what does this mean?). If published, this will include your full peer review and any attached files.

Reviewer #1: No

Reviewer #2: No

Reviewer #3: No

---

## [Author Response · Author response to Decision Letter 0]

4 Aug 2020

Point by point response to reviewer’s comments

Manuscript ID: Ms. [PONE-D-19-30288R1] 

PONE-D-19-30288

Recurrence rate of preterm birth and associated factors among women who delivered at Kilimanjaro Christian Medical centre in Northern Tanzania: a registry based cohort study

Journal: Plose One

Subject: Author's response to reviewer’s comments

Dear Editor, 

Thanks very much for the invitation to resubmit our manuscript entitled: “Recurrence rate of preterm birth and associated factors among women who delivered at Kilimanjaro Christian Medical centre in Northern Tanzania: a registry based cohort study”. We appreciate the reviewers’ useful comments and suggestions to improve our manuscript. We were able to address all the comments and suggestions, and therefore we are expecting that the paper can now be accepted in your journal. Please find our responses are indicated in a red colour. A comments requested have been incorporated in the main text.

Dr. Nathaniel Halide Kalengo

Corresponding author

 

EDITORIAL COMMENTS

1. Please include additional information regarding the survey or questionnaire used in the study and ensure that you have provided sufficient details that others could replicate the analyses. For instance, if you developed a questionnaire as part of this study and it is not under a copyright more restrictive than CC-BY, please include a copy, in both the original language and English, as Supporting Information. If the original language is written in non-Latin characters, for example Amharic, Chinese, or Korean, please use a file format that ensures these characters are visible.

We have included a copy of questionnaire used in this tudy as suggested by the editor.

a) Did participants provide their written or verbal informed consent to participate in this study?

Participants provided oral informed consent, particularly because the interview was administered just after the woman had given birth.

As indicated above, participant provided oral informed consent because interviews were administered just after the woman had given birth. The local Institutional Review Board of KCMC hospital and the National Ethics Committee in Norway approved these procedures when establishing this birth registry.

The KCMC medical birth registry data contains potentially identifying and sensitive patient information. This has also been stipulated by the local Institutional Review Board of KCMC hospital and the National Ethics Committee in Norway when establishing this birth registry. Permission to use the data in this study was made through the Kilimanjaro Christian Medical University College Research and Ethics Review Committee, and received an approval number 2086. The authors do not have the legal right to share the data publicly. All data requests can be sent directly to the Executive Director of the KCMC referral hospital, P. O. Box 3010, Moshi, Tanzania, Email: kcmcadmin@kcmc.ac.tz or through the corresponding author.

Comments to the Author

Reviewer #2: Overall:

General comment:

There are few grammatical errors which need to be corrected.

We thank the reviewer for important suggestion. We have now corrected all the grammatical errors in the manuscript

Title: Well written, clear, and in line with the objectives. It explains what has been researched.

We do appreciate for the complement.

Abstract: The abstract is provided and it contains all the important elements of the study.

We do appreciate for the complement.

Introduction:

The subject has been well researched. There is mixed information about the concept of risk and recurrence in second and third paragraphs, which I feel should be separated.

Last but one paragraph shows that the “study will be done”. This needs to be written in past tense.

We thank the reviewer for these important observations. We are also very sorry for this confusion. The risk of preterm birth in the subsequent pregnancy and recurrence of preterm birth mean the same in this context.

The phrase ‘study will be done’ has been corrected, as per reviewer suggestion 

Methods

The methods have been explained, and have provided answers to raised questions, but do not explain whether this was a prospective or retrospective study. This could explain the data that was missing.

The correction has been done as per reviewer’s comment. This was a retrospective study.

The category for the late preterm birth as explained it the first paragraph of variable definitions is “<37 weeks of gestation age, which means 36 completed weeks. This means there are women who were missed who were in the 37th week and had not attained 37 completed weeks

No women were missed in this categorization since the category < 37 weeks means 36 weeks plus 6 days and not just 36 completed weeks. 

Ethical considerations:

It was ethical to carry out this study as it caused no reported harm to the participants.

Ethical issues were well treated as women gave informed consent, even though it is not explained whether there were women who did not give consent and what happened to them.

There were women who did not give consent and were not included in the study, and this will be corrected in the main document as per reviewer’s observation.

Results

The results section is provided, and explains the results well. There is too much text that explains the results while the tables are provided. The narrative part can concentrate on highlighting only important findings in the table. Some sentences are not very clear, and few are hanging.

The results section was paraphrased and some hanging sentences were taken care of as per reviewer’s suggestion.

Discussion

This is provided and has discussed the results of their study but it does not discuss very well the results and link them well with other previous studies.

We understand reviewer’s concern. We have tried to communicate what were the findings and compare them to other studies in the same settings and in different settings in order to come up with relevant solutions in our setting.

Conclusion

Conclusions have been given. It’s important to limit the conclusions to only what has been studied.

This has been taken care of.

Reviewer #3: 

Thank you for the opportunity to review this manuscript of an important topic in low income country setting.

Here are my comments to improve the manuscript.

ABSTRACT:

The conclusion misses some of the important variables that were significant Need to mention the numerous factors associated with the recurrence.

The important variables have now been mentioned in this section

INTRODUCTION

Please put the long form of KCMC on page 4 paragraph 1 because it is used for the first time.

Response: Done

Write Sustainable Development Goals and not sustainable development goals on page 4 paragraph 2

Response: Done

METHODOLOGY

Study population and study sample: It is good practice to show power calculation to see whether the sample size in the database is enough to depict the outcome intended.

The convenient sampling method was used and women with more than 1 delivery in the birth registry was included in the study. For a cohort study design with around 5000 participants with more than on point data information is enough number to depict the outcome of interest.

Why were the referred cases with medical complications from rural area excluded. Some could have been referred due to conditions that needed medically induced delivery in a tertiary institution.

The referrals with complications were excluded because they would over estimate the outcome of interest.

Variable definitions: There are other important conditions associated with PTb such as diabetes mellitus, chronic hypertension. But these were not factored in. Can you explain if these conditions are not captured in the database.

These factors are in the database, but they were not statistically significant associated with outcome of interest at crude analysis and were thus factored out. 

Data Analysis: You should add a description about risk ratio in the text.

This was added as per reviewer’s suggestion.

Ethical considerations: Write long form for KCMUCo

This has been addressed as per reviewer’s suggestion.

RESULTS

Page 6 last paragraph: "Through spontaneous vaginal delivery (68.15%)". This sentence is hanging it is not clear.

This has been taken care of as per reviewer’s suggestion.

Table 1: Please check the age range 15 -25 and 25 -35. It seems 25 years are included into the two groups which is misleading in my opinion

In the do file analysis the cut off point was 24.999, therefore the greater than or less than signs will be used to clarify this.

Table2: Please rearrange the IPI starting with <24 then 24-29 etc. 

Response: We understand the concern of a reviewer, but an IPI of 24 to 59 months is the optimal IPI as per WHO. This was put at the top as it is termed as baseline group, while less than this is short IPI and greater than this is long IPI.

On parity how do you have primigravida if the inclusion criteria was two or more singleton deliveries: 

Response: We have primigravida because among deliveries picked with more than 2 records in the database, some of them have their primigravida record, second and third deliveries. i.e. these consecutive deliveries include some first pregnancy to some participants while others may have a record of second to fourth pregnancy.

The number of antenatal visits: what about those with four visits it seems they are missing here.

The correction done as per reviewer’s suggestion, now it read 4 and more visits instead of more than four visits versus less than four.

Table 3: A bit confusing as the row numbers add more than the total. May be you need to remove the column with TOTAL N.

Total N= Term deliveries (5405) in previous pregnancy + preterm deliveries (541) in the previous delivery making a total of 5946. The numbers in the rows may be added up to the term and preterm numbers in the current pregnancy, after that it is the clinical classification of PTB among preterm deliveries in the current pregnancy.

Recurrence of preterm birth; I would suggest you put a numerator and denominator for there overall recurrent rate of 24.4% as it not very clear from table 3.

The numerator for this recurrent rate of 24.4% is preterm deliveries in the current pregnancy (132) while the denominator is preterm delivery in the previous pregnancy (541).

DISCUSSION

Page 8 last paragraph: Why do you think it has increased. Is because of referral cases or because of the population increase or because of increase in medically induced deliveries?

This has been added in the text as per reviewer’s suggestion.

Pg 9 Paragraph 1. You can not really make comparison using the meta analysis in my opinion. Rather may be look at the individual papers in meta analysis to make a comparison.

We understand the reviewer concern. This was the best information available, I think the meta-analysis is a summary of many finding, and to my opinion it is ok to make comparison with the findings from my study.

Pg 9 Paragraph 3:For the medically indicated preterm of 44.6%, Probably should say that for medically induced, preterm care should be improved to make sure the premature babies are taken care properly rather than need to be avoided as you can not avoid if delivery is indicated before term.

Medically indicated preterm birth are those which occur due to medical condition which make us deliver the baby earlier, those are the conditions to be addressed in order to push the gestation age to term or close to term.

Paragraph 3 page 10: I think family planning is important but those with long IPI probably were using FP or thee was delay because of other reasons. I think here what is important is to educate the women on the risks of long IPI on preterm birth during the antenatal period.

This observation is well noted and put to action as per reviewer’s suggestion ________________________________________

---

## [Decision Letter · Decision Letter 1]

31 Aug 2020

Recurrence rate of preterm birth and associated factors among women who delivered at Kilimanjaro Christian Medical centre in Northern Tanzania: a registry based cohort study

PONE-D-19-30288R1

Dear Dr. Kalengo,

We’re pleased to inform you that your manuscript has been judged scientifically suitable for publication and will be formally accepted for publication once it meets all outstanding technical requirements.

Kind regards,

Joel Msafiri Francis, MD, MS, PhD

Academic Editor

PLOS ONE

Additional Editor Comments (optional):

Reviewers' comments:

Reviewer's Responses to Questions

**Comments to the Author**

1. If the authors have adequately addressed your comments raised in a previous round of review and you feel that this manuscript is now acceptable for publication, you may indicate that here to bypass the “Comments to the Author” section, enter your conflict of interest statement in the “Confidential to Editor” section, and submit your "Accept" recommendation.

Reviewer #2: All comments have been addressed

Reviewer #3: All comments have been addressed

2. Is the manuscript technically sound, and do the data support the conclusions?

Reviewer #2: (No Response)

Reviewer #3: Yes

3. Has the statistical analysis been performed appropriately and rigorously? 

Reviewer #2: (No Response)

Reviewer #3: Yes

4. Have the authors made all data underlying the findings in their manuscript fully available?

Reviewer #2: (No Response)

Reviewer #3: No

5. Is the manuscript presented in an intelligible fashion and written in standard English?

Reviewer #2: (No Response)

Reviewer #3: Yes

6. Review Comments to the Author

Reviewer #2: (No Response)

Reviewer #3: (No Response)

7. PLOS authors have the option to publish the peer review history of their article (what does this mean?). If published, this will include your full peer review and any attached files.

Reviewer #2: No

Reviewer #3: **Yes: **Furaha August

---

## [Editor Report · Acceptance letter]

4 Sep 2020

PONE-D-19-30288R1 

Recurrence rate of preterm birth and associated factors among women who delivered at Kilimanjaro Christian Medical centre in Northern Tanzania: a registry based cohort study 

Dear Dr. Kalengo:

I'm pleased to inform you that your manuscript has been deemed suitable for publication in PLOS ONE. Congratulations! Your manuscript is now with our production department. 

Kind regards, 

on behalf of

Dr. Joel Msafiri Francis 

Academic Editor

PLOS ONE